# Discovering Hidden Algebraic Structures via Transformers with Rank-Aware Beam GRPO

**Gio Huh**[*]                                                                            *ghuh@caltech.edu*
*Department of Computing and Mathematical Sciences*
*California Institute of Technology*

**Jaeha Lee**[*]                                                                        *jaeha@caltech.edu*
*Division of Physics, Mathematics and Astronomy*
*California Institute of Technology*

**Ning Su**                                                                *ningsu@mail.tsinghua.edu.cn*
*Yau Mathematical Sciences Center*
*Tsinghua University*

**Tony Yue Yu**                                                                              *tyy@caltech.edu*
*Division of Physics, Mathematics and Astronomy*
*California Institute of Technology*

**Reviewed on OpenReview:** *https://openreview.net/forum?id=Vxf8QDIA6Z*

## Abstract

Recent efforts have extended the capabilities of transformers in logical reasoning and symbolic computations. In this work, we investigate their capacity for functional decomposition, focusing on the challenging algebraic task of multivariate polynomial decomposition. This problem, with widespread applications in science and engineering, is proved to be NP-hard, and demands both precision and insight. Our contributions are threefold: First, we develop a synthetic data generation pipeline providing fine-grained control over problem complexity. Second, we train transformer models via supervised learning and evaluate them across four key dimensions involving scaling behavior and generalizability. Third, we propose Beam Grouped Relative Policy Optimization (BGRPO), a rank-aware reinforcement learning method suitable for hard algebraic problems. Fine-tuning with rank-aware BGRPO improves beam-search accuracy by 33–37 percentage points over the SFT initialization and by 1.7–3.7 points over vanilla GRPO, with non-overlapping $\pm 1\sigma$ bands at every scale. After RL, even greedy decoding surpasses the SFT model's best beam-search score by 23.8–25.8 percentage points at every scale. Additionally, our model demonstrates competitive performance with Mathematica's FullSimplify on leaf count in various cases.

## 1 Introduction

Transformers, initially developed for natural language processing (Vaswani et al., 2017), have shown remarkable versatility across diverse domains such as vision (Dosovitskiy et al., 2020) and protein folding (Jumper et al., 2021). More recently, their applications in formal reasoning, symbolic mathematics and algorithmic tasks start to gain traction. Several works have demonstrated transformer-based architectures' ability to tackle highly structured problems, including theorem proving (Polu & Sutskever, 2020; Trinh et al., 2024), integration (Lample & Charton, 2020), matrix multiplication (Fawzi et al., 2022) and equation solving (Drori et al., 2022).

---

[*]Equal contribution.

In this work, we investigate the transformer's capacity for functional decomposition, i.e. decomposing a complex function as the composition of simpler sub-functions. In contrast to step-by-step logical deduction, or pattern recognition in data analysis, functional decomposition poses significant new challenges to the transformer, because the forms of the sub-functions that we try to discover can be totally hidden or obscured in the final compact form of the original function. Furthermore, it demands exact precision, as correctness is judged by strict symbolic equality, so a single sign flip in one coefficient invalidates the entire answer. This is what sets decomposition apart from more forgiving classification settings, which are typically scored with soft metrics such as top-$k$ accuracy, partial credit, or probability mass on the correct class. Here, by contrast, a full token sequence must match exactly, so the set of correct solutions is sparse.

Beyond its theoretical interest, functional decomposition has ubiquitous applications in software engineering (Tempero et al., 2024), systems biology (Mori et al., 2023), mechanical design (She et al., 2024), systems engineering (Hernandez et al., 2024) and digital logic design (Adamski et al., 2005; Lin et al., 2008), where capturing hidden substructures within high-dimensional functions leads to more tractable and efficient models. However, identifying a function's latent compositional structure requires models to look past token-level co-occurrence patterns that do not reflect the underlying algebraic structure, attending instead to the symmetries and invariants that define the composition.

A particularly rich case of functional decomposition arises in multivariate polynomial functions. The polynomial decomposition problem over a ring $k$ seeks to decompose a given polynomial $f \in k[x_1, \ldots, x_n]$ into polynomials $g \in k[y_1, \ldots, y_m]$ and $h_1, \ldots, h_m \in k[x_1, \ldots, x_n]$ such that

$$f(x_1, \ldots, x_n) = g\big(h_1(x_1, \ldots, x_n), \ldots, h_m(x_1, \ldots, x_n)\big). \tag{1}$$

It has wide-ranging applications from cryptography (Patarin & Goubin, 1997) to dynamical modeling (Dang & Testylier, 2012), signal processing (Demirtas et al., 2012) and robotics (Elias & Wen, 2025; Manocha & Canny, 1992).

The multivariate polynomial decomposition problem has been proved to be NP-hard by Dickerson (Dickerson, 1987; 1993), although efficient algorithms for various special cases are discussed in (Von Zur Gathen et al., 2003; Von Zur Gathen, 1990a;b; Faugère & Perret, 2009a;b; Zhao et al., 2012). To illustrate the difficulty of the problem for the models, let us consider the following expression

$$\begin{aligned}
f =& 2a_1^3b_1^3 + 25a_1^2b_1^2 + 6a_1^2a_2b_2b_1^2 + 6a_1^2a_3b_3b_1^2 + 6a_1a_2^2b_2^2b_1 + 6a_1a_3^2b_3^2b_1 \\
&+ 96a_1b_1 + 50a_1a_2b_2b_1 + 50a_1a_3b_3b_1 + 12a_1a_2a_3b_2b_3b_1 + 2a_2^3b_2^3 + 2a_3^3b_3^3 \\
&+ 25a_2^2b_2^2 + 25a_3^2b_3^2 + 6a_2a_3^2b_2b_3^2 + 96a_2b_2 + 6a_2^2a_3b_2^2b_3 + 96a_3b_3 \\
&+ 50a_2a_3b_2b_3 + 128
\end{aligned}$$

It has a hidden $O(3)$-symmetry, which can be revealed by decomposing $f = g \circ h$, with $g(y) = y^2 + 2(4 + y)^3$ and $h = a_1b_1 + a_2b_2 + a_3b_3$. This is a highly nontrivial task to identify the inner function $h$ directly from the expanded form of $f$, as its structure becomes completely obscured after polynomial substitution, expansion and simplification. Even in this relatively constrained case where $g$ is univariate, discovering the decomposition requires recognizing non-linear latent patterns across dozens of terms. When $g$ becomes multivariate, the complexity increases substantially, making the problem even more challenging.

To tackle the polynomial decomposition problem, we develop a systematic approach with four key components. First, we create a backward synthetic data generation pipeline that allows fine-grained control over polynomial complexity involving range of coefficients, degree, and number of variables. Second, we train lightweight transformer models on these synthetic datasets using supervised learning and analyze how performance scales across four axes (performance complexity scaling, architecture scaling, distribution adaptation, search strategy analysis). Third, we discover that both multi-sampling and greedy search methods struggle with the sparse solution space of the polynomial decomposition problem, and we implement a beam search strategy to effectively extract the models' capabilities. Finally, we develop a rank-aware variant of the Grouped Relative Policy Optimization (GRPO) reinforcement learning algorithm, which encodes rank information directly in the reward function.

Our study makes the following contributions to neural approaches for polynomial decomposition. First, our backward data generation pipeline enables targeted training across varying levels of decomposition

difficulty. Second, our evaluation across four axes ($\mathcal{D}_1$–$\mathcal{D}_4$) establishes the first baselines for transformers on polynomial decomposition. Third, our rank-aware Beam Grouped Relative Policy Optimization (BGRPO) lifts beam-search accuracy by 33–37 percentage points over the SFT initialization and by 1.7–3.7 points over vanilla GRPO (non-overlapping $\pm 1\sigma$ bands at every scale); after BGRPO, greedy decoding alone exceeds the SFT model's best beam-search score by 23.8–25.8 points at every scale. Additionally, our model demonstrates competitive performance with Mathematica's FullSimplify on leaf count in various cases. This shows that neural models can complement and extend classical symbolic computation.

## 2 Method

### 2.1 Backward Synthetic Data Generation

We generate synthetic data for supervised learning using a backward approach, starting from the decomposed form. First, we generate the inner functions ($h_1, \ldots, h_m$ in Eq. (1)) and the outer function ($g$ in Eq. (1)) with random monomial terms of bounded degree and random coefficients within a given range. Then, we obtain the composed function ($f$ in Eq. (1)) via substitution, expansion, and term collection. See Appendix A for the detailed algorithm. For each generated instance, we create a training pair consisting of the expanded polynomial $f$ as input and its decomposed components $\{g, h_1, \ldots, h_{v_{\text{outer}}}\}$ as the target output. The model is trained to minimize the standard negative log-likelihood loss function.

Our synthetic data generation process provides fine-grained control over problem complexity through eight parameters: $C_{\text{inner}}$ (coefficient range for inner polynomials), $d_{\text{inner}}$ (maximum degree of inner polynomials), $v_{\text{inner}}$ (number of variables in inner polynomials), $t_{\text{inner}}$ (maximum number of terms in inner polynomials), and similarly $C_{\text{outer}}$, $d_{\text{outer}}$, $v_{\text{outer}}$, and $t_{\text{outer}}$ for the outer polynomial.

### 2.2 Beam Search

Beam search is a breadth-first search algorithm that approximates optimal decoding by keeping track of the $w$ most probable sequences at each step (Freitag & Al-Onaizan, 2017). For each of the $w$ current sequences, the algorithm considers the top-$w$ token extensions per sequence. These $w^2$ candidate continuations are then ranked by the sum of log probabilities of all tokens in the sequence, and only the top-$w$ sequences with the highest cumulative log probability are retained for the next step. In this paper, we refer to $w$ as the beam width, and to the position (1st, 2nd, etc.) of an output in the final beam as its rank.

Our analysis across all model outputs identified a specific error pattern in polynomial decomposition. The model achieves approximately 90% accuracy for predicting non-sign tokens (operators, numbers, variables), but exhibits near-random performance for deciding between positive and negative signs. This creates a unique inference challenge where exploration needs to be constrained for high-confidence structural elements while simultaneously expanded for uncertain sign choices.

Beam search is particularly well-suited for this situation as it maintains the high-confidence structural backbone while systematically exploring variations in the uncertain components. Our experiments demonstrate that beam search significantly outperforms greedy decoding and random sampling for polynomial decomposition tasks. See Appendix C for a detailed error analysis and an explanation of beam search effectiveness for this task.

### 2.3 BGRPO: A Rank-Aware Reinforcement Learning Method

Supervised training alone leaves the model near-random on a small subset of tokens (the $\pm$ signs in our case; see Section 2.2), which beam search at inference partially compensates for. To raise accuracy at the policy level rather than only at decoding time, we introduce Beam Grouped Relative Policy Optimization (BGRPO), a reinforcement learning method that extends GRPO by generating its group of outputs through beam search and by rewarding correct answers in proportion to their rank in the beam.

Reinforcement learning enables models to explore solution spaces more effectively than supervised learning alone, enhancing the model's capabilities by addressing specific weaknesses through a reward mechanism. This

approach encourages correct answers while discouraging incorrect ones based on an advantage function—the difference between a solution's reward and a baseline reward. Group Relative Policy Optimization (GRPO) (Shao et al., 2024) estimates this baseline for each question by sampling a group of outputs, and has shown promising results for reinforcement learning in language generation tasks due to its sample efficiency and stability (DeepSeek-AI, 2025). We chose GRPO over traditional Proximal Policy Optimization (PPO) (Schulman et al., 2017) because it eliminates the need for a separate value network or reward model, reducing training complexity while improving stability, and its group-wise baseline calculation naturally fits tasks with a clear binary reward structure like polynomial decomposition.

Our proposed Beam Grouped Relative Policy Optimization (BGRPO) extends this approach by using beam search rather than independent sampling for generating the group of outputs. While this significantly alters the distribution of outputs, making their average reward less suitable as a traditional baseline, it still provides valid training signals by reinforcing correct answers and penalizing incorrect ones. BGRPO is particularly effective for our task because beam search generates outputs with identical structure that differ only in the confusing elements (signs), creating a focused learning signal.

Additionally, BGRPO incorporates rank information directly into the reward function by applying an exponential decay factor based on the position in the beam. This incentivizes correct answers to appear at earlier positions in the beam search, effectively pushing correct solutions toward the top of the beam ranking.

**Training Objective**  For a prompt $x$, let $\mathcal{B}(x) = \{y_1, \ldots, y_w\}$ be the set of beam search outputs with beam width $w$ generated by the old policy $\pi_{\theta_{\text{old}}}$, where each $y_i$ has length $|y_i|$ tokens. Each output sequence $y_i$ receives a sequence-level reward $r_i$, where $r_i = 0$ for incorrect polynomial decomposition and $r_i = 1$ for correct decomposition. In BGRPO, we incorporate rank information by scaling the reward for correct decompositions by $e^{-\text{rank}/b}$. The decay base $b$ is a hyperparameter independent of the beam width $w$; we ablate it in Section 4.5. We optimize the policy model $\pi_\theta$ for $\mu$ inner gradient steps by maximizing the following token-level surrogate, following the GRPO formulation of Shao et al. (2024):

$$\mathcal{J}_{\text{BGRPO}}(\theta) = \frac{1}{\sum_{i=1}^{w} |y_i|} \sum_{i=1}^{w} \sum_{t=1}^{|y_i|} \left( \min(\rho_{i,t} A_i, \text{ clip}(\rho_{i,t}, 1 - \varepsilon, 1 + \varepsilon) A_i) - \beta \, \mathbb{D}_{\text{KL}}(\pi_\theta \| \pi_{\text{ref}})_{i,t} \right), \tag{2}$$

where $\rho_{i,t} = \pi_\theta(y_{i,t} \mid x, y_{i,<t}) / \pi_{\theta_{\text{old}}}(y_{i,t} \mid x, y_{i,<t})$ is the per-token importance ratio, $\varepsilon$ is the PPO clipping parameter, and $\beta$ controls the KL regularization. The per-token KL is approximated as

$$\mathbb{D}_{\text{KL}}(\pi_\theta \| \pi_{\text{ref}})_{i,t} = \frac{\pi_{\text{ref}}(y_{i,t} \mid x, y_{i,<t})}{\pi_\theta(y_{i,t} \mid x, y_{i,<t})} - \log \frac{\pi_{\text{ref}}(y_{i,t} \mid x, y_{i,<t})}{\pi_\theta(y_{i,t} \mid x, y_{i,<t})} - 1. \tag{3}$$

Here, $\pi_{\text{ref}}$ is the reference policy, fixed to the SFT initialization before BGRPO training. The advantage $A_i$ is shared across all tokens of beam $y_i$ and is computed without standard-deviation normalization as $A_i = r_i - \bar{r}$, where $\bar{r} = \frac{1}{w} \sum_{j=1}^{w} r_j$, following Liu et al. (2025).

## 3 Experimental Setup

### 3.1 Evaluation Axes

To systematically analyze our models' capabilities for the polynomial decomposition problem, we consider four key evaluation dimensions.

**Problem Complexity Scaling ($\mathcal{D}_1$).** We analyze how the model performance varies with respect to changes in the complexity parameters for synthetic data generation. We vary the number of variables $v_{\text{inner}}, v_{\text{outer}}$, and the maximum degrees $d_{\text{inner}}, d_{\text{outer}}$ for both the inner and outer polynomials.

**Architecture Scaling ($\mathcal{D}_2$).** We investigate how model performance scales with key architectural hyperparameters of the transformer. In particular, we measure $\mathcal{P}(M(d, l, a))$, the performance of models with embedding dimension $d$, number of layers $l$, and number of attention heads $a$. Our goal is to characterize how these hyperparameters influence model capabilities.

**Distribution Adaptation ($\mathcal{D}_3$).** A practical challenge in applying transformers to symbolic computation is their sensitivity to the numerical ranges present in the training data. For example, models trained on specific

coefficient ranges tend to struggle with polynomials outside these ranges. On the other hand, we found that models can rapidly adapt to new coefficient distributions with minimal additional training, suggesting that they manage to learn generalizable pattern recognition rather than merely memorizing specific numerical relationships.

To quantify the model's ability to transfer its polynomial decomposition skills to numerically distinct but structurally identical problems, we prepare the model $M^n_{C_1 \to C_2}$. This model is initially trained on 1M polynomial decomposition examples with $C_{\text{outer}} = C_1$ and then fine-tuned with $n$ examples with $C_{\text{outer}} = C_2$ where $C_1 \cap C_2 = \emptyset$. We measure the performance of model $M^n_{C_1 \to C_2}$ on a test set of polynomial decomposition problems with $C_{\text{outer}} = C_2$:

$$\mathcal{G}(n) = \mathcal{P}\big(M^n_{C_1 \to C_2}, \text{ test set with } C_{\text{outer}} = C_2\big) \tag{4}$$

**Search Strategy Analysis ($\mathcal{D}_4$).** We investigate how beam search enhances model performance on polynomial decomposition tasks, analyzing its effectiveness across different model architectures and levels of problem complexity.

### 3.2 Synthetic Dataset Setup

For the axis $\mathcal{D}_1$ of the problem complexity scaling, we first examine degree scaling by training a model on 2M polynomial decomposition examples with different inner and outer degrees as described in Table 1. We then evaluate this model on separate test datasets with the same configuration parameters, each corresponding to one of nine different $(d_{\text{inner}}, d_{\text{outer}})$ pairs to assess performance across varying problem complexities.

For the second part of the $\mathcal{D}_1$ axis, we train a model for each combination of $v_{\text{inner}}$ and $v_{\text{outer}}$ varying from 2 to 4 while fixing the other parameter at 3. For each combination, we use 1M examples to train the model.

For the axis $\mathcal{D}_2$ of architecture scaling, we train multiple models with varying architectural configurations, all using the same dataset of 2M examples with polynomial parameters as described in Table 1.

For the axis $\mathcal{D}_3$ of distribution adaptation, we train initial models on 1M examples with $C_{\text{outer}} = C_1 = [-5, 5]$ and then adapt them to examples with $C_{\text{outer}} = C_2 = [-10, -6] \cup [6, 10]$. Other parameters are the same across both datasets as described in Table 1.

For the second part of $\mathcal{D}_1$ (Variable Scaling) and $\mathcal{D}_2$, we set $t_{\text{inner}} = t_{\text{outer}} = 3$ to prevent expressions from becoming too long. We describe our tokenization in Appendix B.

Table 1: Synthetic Dataset Configuration Across Evaluation Axes

| Evaluation Axis | Inner Coeff. | Outer Coeff. | Inner Degrees | Outer Degrees | Inner Vars | Outer Vars |
|---|---|---|---|---|---|---|
| $\mathcal{D}_1$ (Degree Scaling) | $[-20, 20]$ | $[-20, 20]$ | $\{2, 3, 4\}$ | $\{2, 3, 4\}$ | 1 | 1 |
| $\mathcal{D}_1$ (Variable Scaling) | $[-5, 5]$ | $[-5, 5]$ | 3 | 3 | $\{2, 3, 4\}$ | $\{2, 3, 4\}$ |
| $\mathcal{D}_2$ (Architecture) | $[-5, 5]$ | $[-5, 5]$ | 3 | 3 | 3 | 3 |
| $\mathcal{D}_3$ (Adaptation) | $[-20, 20]$ | $C_1 = [-5, 5]$ | $\{1, 2\}$ | $\{1, 2, 3, 4\}$ | 1 | 1 |
| | $[-20, 20]$ | $C_2 = [-10, -6] \cup [6, 10]$ | $\{1, 2\}$ | $\{1, 2, 3, 4\}$ | 1 | 1 |

### 3.3 Architecture Configuration

We employ a decoder-only transformer architecture following standard design principles (Vaswani et al., 2017). Table 2 summarizes our task-specific configurations across all experimental axes. For lightweight and effective training, we developed our own model and training pipeline based on `minGPT` (Karpathy, 2020).

### 3.4 Supervised Learning Details

We train our models using the Adam optimizer with an initial learning rate of $6 \times 10^{-4}$, incorporating a 10% warmup period followed by cosine decay. Each configuration initially trains on 1M instances, with

Table 2: Transformer Model Configuration Across Experiments

| Experiment | Context Window | Embedding Dim. | Layers | Heads |
|---|---|---|---|---|
| $\mathcal{D}_1$ (Degree Scaling) | 256 | 512 | 6 | 8 |
| $\mathcal{D}_1$ (Variable Scaling) | 850 | 512 | 6 | 8 |
| $\mathcal{D}_2$ (Architecture) | 850 | $\{256, 512, 768\}$ | $\{4, 6\}$ | 8 |
| $\mathcal{D}_2$ (Attention Heads) | 850 | 512 | 6 | $\{4, 8, 16\}$ |
| $\mathcal{D}_3$ (Distribution) | 256 | 512 | 4 | 8 |

*Common settings: GELU activation, learned positional embeddings, multi-head attention with causal masking, MLP hidden dimension = 4× embedding dimension.*

additional 1M training examples added incrementally until performance saturation. We use a batch size of 200 throughout training. We train models with enough epochs until it saturates with the given dataset.

### 3.5 BGRPO Implementation

For the BGRPO reinforcement learning phase we generate candidate solutions using beam search with a width of 32 and temperature of 1.0. The rank-aware reward uses decay base $b = 20$ as the default; the advantage is computed without standard-deviation normalization as $A_i = r_i - \bar{r}$, matching the specification in Section 2 and following Liu et al. (2025). We set the PPO clipping parameter $\varepsilon$ to 0.2 and the KL divergence coefficient $\beta$ to 0.01, with a learning rate of $1 \times 10^{-5}$. Each outer training step samples 8 distinct polynomial decomposition problems from a held-in pool of 200 non-repeating problems, disjoint from the evaluation set, and applies 5 inner gradient steps. We train for 420 outer steps and report results averaged across 3 random seeds (148, 1, 2). Evaluation uses greedy decoding on 200 held-out problems and beam search of width 30 on the same 200 problems, drawn from the multi-variable test set used in $\mathcal{D}_2$.

## 4 Experimental Results

### 4.1 Problem Complexity Scaling ($\mathcal{D}_1$)

In the first part of $\mathcal{D}_1$, we examine how model performance varies with the degrees of inner and outer polynomials. The result is shown in Figure 1. We use greedy search for the inference. Regardless of the degrees of the polynomials, our model achieves a remarkable single-output accuracy. Notably, when using beam search with a width of 10, the model's accuracy reaches 100% for these configurations.

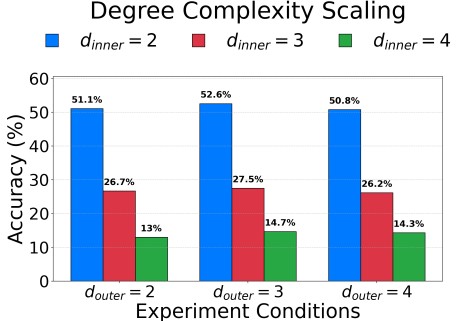
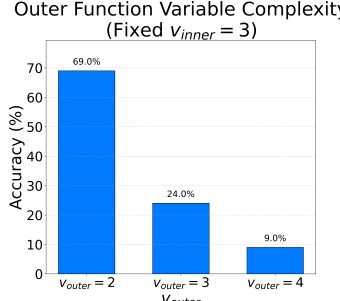
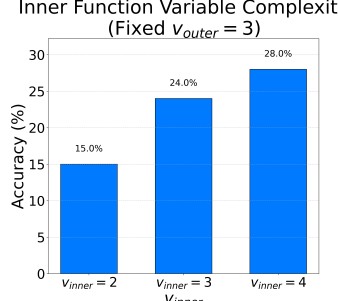

Figure 1: Performance across different $d_{\text{inner}}, d_{\text{outer}}$.[1]

Figure 2: Performance across different $v_{\text{outer}}$

Figure 3: Performance across different $v_{\text{inner}}$

Our analysis reveals that performance remains invariant to increases in the outer polynomial's degree, while decreasing when the inner polynomial's degree increases. This demonstrates that the transformer's

---

[1]Results for $\mathcal{D}_1$ and $\mathcal{D}_2$ are from single training runs due to compute constraints.

decomposition capability is primarily limited by the complexity of the inner polynomial rather than that of the outer polynomial.

In the second part of $\mathcal{D}_1$, we investigate how the performance scales with $v_{\text{inner}}$ and $v_{\text{outer}}$, the number of variables in the inner and outer polynomials. Figures 2 and 3 present these results.

Given the challenging nature of multivariate polynomial decomposition, we evaluate the model's performance using beam search with a width of 30, considering a prediction correct if at least one of the 30 candidate outputs is correct decomposition.

Our results reveal that performance decreases dramatically as $v_{\text{outer}}$ increases, yet counter-intuitively improves as $v_{\text{inner}}$ increases. This observation aligns with the following heuristic understanding that higher $v_{\text{outer}}$ creates an information bottleneck, requiring the model to simultaneously resolve multiple interdependent inner functions. In contrast, higher $v_{\text{inner}}$ provides more dimensions of input variation with additional structural indicators that can guide the decomposition process.

## 4.2 Architecture Scaling ($\mathcal{D}_2$)

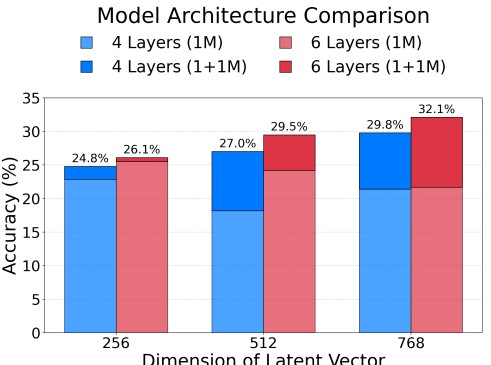

Figure 4: Accuracies on different number of layer and dimension.

In $\mathcal{D}_2$, we examine how model performance varies with architectural parameters: embedding dimension, number of layers, and number of attention heads. When varying the number of heads, we maintain a constant total embedding dimension, meaning that models with more heads have smaller per-head embedding dimensions. We use the dataset described in Section 3.2 and evaluate using beam search with a width of 30.

Figure 4 reveals the scaling behavior (Kaplan et al., 2020) of transformer architectures on polynomial decomposition. As model capacity increases through higher embedding dimensions and additional layers, performance consistently improves.

Notably, our results demonstrate the presence of a data-dependent scaling threshold. With limited training data (1M examples), larger models initially underperform their simpler counterparts, particularly evident in the 6-layer configurations with higher embedding dimensions. However, this pattern reverses completely with additional training data, confirming that larger models possess superior capacity for mathematical pattern recognition when provided with sufficient examples to leverage their parametric advantage.

In $\mathcal{D}_2$, we also examine model performance with different numbers of attention heads. Our experiments reveal that increasing the number of attention heads while maintaining constant total embedding dimension leads to progressively deteriorating performance on polynomial decomposition tasks. Models with 4 heads achieved 32.0% accuracy, while those with 8 and 16 heads reached only 28.0% and 25.0% accuracy, respectively. This suggests that for our specific task of mathematical pattern recognition, fewer, more expressive attention heads with larger per-head dimensions provide better performance than numerous specialized heads with smaller dimensions.

## 4.3 Distribution Adaptation ($\mathcal{D}_3$)

We evaluate $\mathcal{G}(n)$ as defined in Eq. 4, which measures how quickly models adapt to new coefficient distributions as a function of adaptation sample size $n$. For this experiment, we train a model with 4 layers and 512 embedding dimension on the dataset described in Section 3.2. The initial training used 1M examples with outer polynomial coefficient range $C_1$, followed by fine-tuning on $n$ examples with coefficient range $C_2$ for a single epoch. We report the variance in accuracy based on three independent trials.

Models trained exclusively on the first dataset achieve only 5.67% accuracy on the new distribution, despite reaching nearly 100% accuracy on the original distribution. Figure 5 illustrates how performance recovers during adaptation. Notably, despite using only $\approx 2\%$ of the original training data size, the model rapidly

recovers its accuracy from single digits to over 90%. This rapid adaptation indicates successful transfer learning, suggesting that the model develops a general mathematical understanding of polynomial substructures rather than memorizing specific numerical relationships.

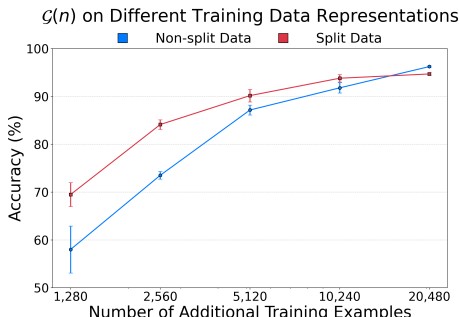

Figure 5: Performance recovery when adapting to a new coefficient distribution

We further investigate whether alternative data representations could enhance this adaptation capability. We propose "split" representation of polynomials, where we randomly select terms from the expanded form and split their coefficients. For example:

$$f_{\text{non-split}}(a) = -63 + 23a - 71a^2 - 11a^3 - 14a^4 - 12a^5 - 2a^6$$
$$f_{\text{split}}(a) = -63 + 23a - 4a^2 - 67a^2 - 8a^3 - 3a^3$$
$$- 7a^4 - 7a^4 - 12a^5 - a^6 - a^6 \quad (5)$$

In Figure 5, the red line demonstrates $\mathcal{G}(n)$ of the model trained on data with both normal and split representation. Models trained on this mixed data including split representation demonstrate significantly faster adaptation, requiring only 70% of the additional training examples to reach equivalent performance on the new distribution.

This enhanced generalization likely stems from the model being forced to recognize mathematically equivalent but differently represented polynomials, compelling it to develop a deeper understanding of polynomial structure rather than memorizing specific patterns.

## 4.4 Search Strategy Analysis ($\mathcal{D}_4$)

We evaluate how search strategies impact model performance on polynomial decomposition, comparing greedy decoding against beam search across model scales and problem complexities. Figure 6 and 7 illustrate the accuracy achieved across different beam widths for polynomials with varying numbers of variables.

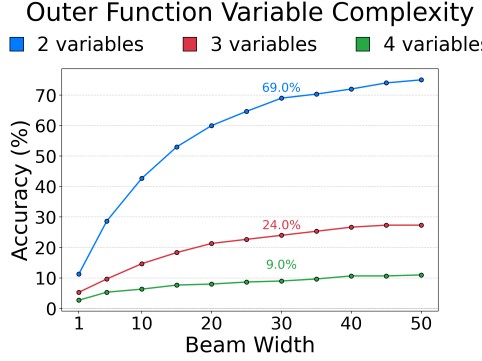

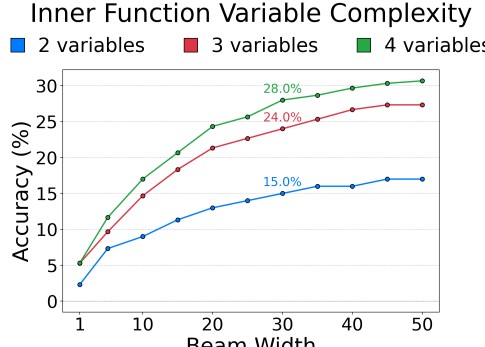

Figure 6: Beam width scaling with varying $v_{\text{outer}}$ ($v_{\text{inner}} = 3$)

Figure 7: Beam width scaling with varying $v_{\text{inner}}$ ($v_{\text{outer}} = 3$)

Our results reveal an unusually large impact of beam search for polynomial decomposition compared to typical NLP tasks. For two-variable polynomials, accuracy improves from 11% with greedy search to 69% with a beam width of 30, a 6.3× gap. This is much larger than in standard neural machine translation, where beam search typically yields BLEU improvements of only 2–4 points (Huang et al., 2018; Ranzato et al., 2016) and most systems show diminishing returns with beam widths beyond 5–10 (Freitag & Al-Onaizan, 2017). We return to this gap in Section 4.5, where rank-aware BGRPO closes it. Greedy decoding alone after BGRPO already exceeds the SFT model's best beam-search score at every scale.

### 4.5 BGRPO Results

Rank-aware BGRPO raises beam@30 accuracy over the SFT initialization by 33–37 accuracy points at every scale. We start from three 6-layer SFT models (embedding dimensions 256, 512, 768) trained on the 2M-example $\mathcal{D}_2$ dataset (Section 3.2) with our released pipeline.[2] On these models, beam@30 climbs from 16.5%, 20.0%, and 20.0% to $53.5 \pm 0.4\%$, $54.5 \pm 0.4\%$, and $53.7 \pm 0.2\%$ respectively. All numbers in this section are mean $\pm$ standard deviation across 3 training seeds, evaluated on 200 held-out multi-variable test problems (beam width 30; see Section 3.5).

Figure 8 traces the full training trajectory for each scale, comparing four conditions at matched compute (420 outer training steps, 8 problems per step): the SFT initialization (dashed gray), vanilla GRPO, BGRPO with a binary reward, and BGRPO with the rank-aware reward at decay base $b = 20$. Lines are mean $\pm 1\sigma$ across 3 training seeds. Two observations stand out. First, vanilla GRPO already closes most of the gap from the SFT initialization to the rank-aware ceiling, reaching $49.8 \pm 1.0\%$, $52.3 \pm 0.2\%$, and $52.0 \pm 0.4\%$ at the three scales. The rank-aware reward adds another 1.7–3.7 accuracy points on top of GRPO, with non-overlapping $\pm 1\sigma$ bands at the converged plateau (outer steps 300–420) at every scale. Second, BGRPO with a flat binary reward is the weakest of the three RL variants at $d = 512$ and $d = 768$, falling 13–15 points below vanilla GRPO. Beam-search rollout offers no advantage over independent sampling when the policy update cannot distinguish among the correct beams it returns. A flat reward assigns the same advantage to every correct beam, discarding the rank signal that the beam rollout collected. This connects to the token-level error structure in Table 6. Beams share the same structural backbone at $\sim$90% per-token accuracy and disagree almost entirely on sign tokens at $\sim$50%. The rank weighting concentrates the learning signal on these divergent sign tokens, pushing the model to resolve sign ambiguities and surface correct patterns toward the top of the beam.

## BGRPO training trajectory at beam@30

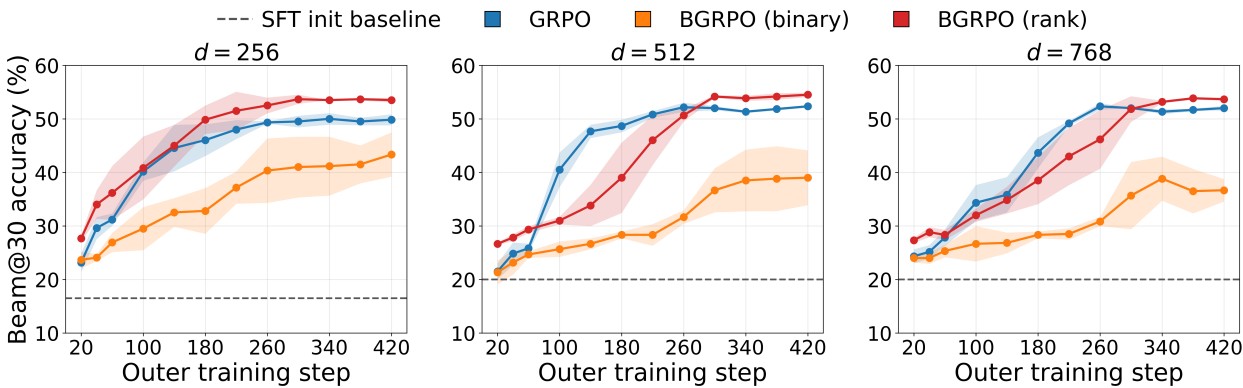

Figure 8: Beam@30 accuracy over outer training step for the three SFT initializations ($d = 256, 512, 768$), comparing vanilla GRPO, BGRPO (binary), and BGRPO (rank-aware, $b = 20$) against the SFT baseline (dashed gray). Shaded bands show $\pm 1\sigma$ across 3 seeds on 200 held-out problems.

**Decay-base ablation.** The rank-aware reward scales correct outputs by $e^{-\text{rank}/b}$ (Section 2); the decay base $b$ is the only free hyperparameter introduced by the rank signal, and BGRPO is highly sensitive to it. Figure 9 sweeps $b \in \{4, 8, 16, 20, 32, 64, 128\}$ on the $d = 256$ initialization at outer step 180 (the common training budget across all sweep runs). The peak-to-trough range spans roughly 10 accuracy points. The curve is unimodal, peaking at $b = 20$ ($49.8 \pm 2.6\%$) and dropping to $41.5 \pm 2.3\%$ at $b = 4$ and $39.8 \pm 2.9\%$ at $b = 128$. The peak sits on the order of the beam width 32, which is the scale at which the exponential can meaningfully distinguish the top of the beam from the bottom. A sharper decay ($b = 4$) concentrates all

---

[2]These initializations were trained separately for the BGRPO study, on the same architecture and 2M dataset as the $\mathcal{D}_2$ models in Figure 4; their absolute SFT accuracies thus differ slightly, and we report BGRPO gains relative to each initialization.

credit on rank 1 and starves the rest of the beam, while a flatter decay ($b = 128$) approaches the binary case where every correct beam looks identical to the policy update. The remainder of this section uses $b = 20$.

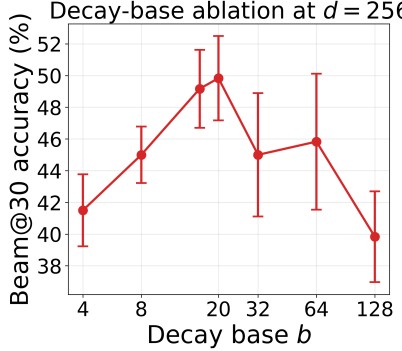

Figure 9: Decay-base ablation on the $d = 256$ initialization at outer step 180. Mean $\pm 1\sigma$ across 3 training seeds.

**Greedy decoding after RL.** After rank-aware BGRPO, greedy decoding alone surpasses the SFT model's best beam-search score at every scale (Table 3). At $d = 256$, the rank-aware model decodes greedily at $42.3 \pm 2.5\%$, exceeding the SFT model's beam@30 score of $16.5\%$ by 25.8 points. The corresponding greedy scores at $d = 512$ and $d = 768$ are $45.7 \pm 1.2\%$ and $43.8 \pm 2.8\%$, against SFT-beam@30 of $20.0\%$ and $20.0\%$ respectively. Notably, the asymmetry between beam search and greedy decoding flips between the two RL variants. Vanilla GRPO is 2.0–3.5 points higher than rank-aware BGRPO at greedy decoding (e.g., $48.7 \pm 0.6\%$ vs $45.7 \pm 1.2\%$ at $d = 512$), while rank-aware BGRPO is 1.7–3.7 points higher at beam@30. This pattern is consistent with the design of the rank decay, which redistributes probability mass into the upper region of the beam rather than strictly to rank 1; the small cost at the very top is recovered and exceeded once more than one beam position is examined.

Table 3: Greedy decoding after rank-aware BGRPO compared with SFT-beam@30. Greedy values are mean $\pm 1\sigma$ across 3 training seeds, evaluated on 200 held-out multi-variable problems.

|  | SFT (beam@30) | rank-aware BGRPO (greedy) | gain |
|---|---|---|---|
| $d = 256$ | 16.5% | $42.3 \pm 2.5\%$ | +25.8 |
| $d = 512$ | 20.0% | $45.7 \pm 1.2\%$ | +25.7 |
| $d = 768$ | 20.0% | $43.8 \pm 2.8\%$ | +23.8 |

**Architecture-scaling compression.** The SFT models exhibit a 3.5-point spread in beam@30 accuracy across the three embedding dimensions (16.5% / 20.0% / 20.0% at $d = 256$ / $d = 512$ / $d = 768$). After rank-aware BGRPO the spread collapses to 1.0 point (53.5% / 54.5% / 53.7%), a factor of roughly $3.5\times$ compression. Greedy decoding shows the same pattern. An SFT greedy spread of 5.0 points (10.0% / 10.0% / 15.0%) becomes 3.4 points (42.3% / 45.7% / 43.8%) after rank-aware BGRPO. The smallest model gains the most under RL. $d = 256$ gains 37 accuracy points at beam@30, against 34.5 and 33.7 points for $d = 512$ and $d = 768$. RL post-training partially substitutes for parameter count on this task; whether the architecture-scaling gap reopens at larger model sizes is left to future work.

### 4.6 Simplification Comparison with Mathematica

While polynomial simplification and polynomial decomposition represent two distinct mathematical objectives, simplification frequently arises as a consequence of decomposition, since decomposed forms generally exhibit reduced algebraic complexity compared to the original expression. In this subsection, we briefly explore the capabilities of our models for this related problem, and benchmark against the most powerful symbolic computation engine Mathematica. Despite our lightweight parameter budgets and the absence of any explicit simplification objective in our training, the models were able to reduce the leaf count (Wolfram Research, Inc., 1988) of complex expressions, with performance competitive with Mathematica's FullSimplify function across the five regimes (see Table 4).

These findings suggest that transformers trained on decomposition can produce simplifications competitive with dedicated symbolic engines, despite never being trained to simplify.

### 4.7 Comparison with Frontier LLMs and Classical Algorithms

Beyond simplification, we compare our model against external baselines on the decomposition task itself. The only available classical algorithm and a panel of frontier LLMs, all evaluated with BGRPO's verifier on the same $(v_O, v_S)$ regimes. We note that neither baseline solves the task.

Table 4: Average leaf count comparison (Beam width = 30). $\Delta$ is transformer minus FullSimplify, so negative values mean fewer leaves. Entries where the transformer matches or beats FullSimplify within one leaf ($\Delta \leq 1$) are bolded.

| Problem Complexity | | Leaf Count (mean) | | |
|---|---|---|---|---|
| $v_O$ | $v_S$ | Transformer | Mathematica | $\Delta$ |
| 2 | 3 | **27.28** | 30.03 | **-2.75** |
| 3 | 3 | 22.85 | **22.12** | **0.73** |
| 4 | 3 | 22.52 | 20.00 | 2.52 |
| 3 | 2 | 17.27 | **17.10** | **0.17** |
| 3 | 4 | **26.04** | 27.56 | **-1.52** |

Table 5: Solve rate by regime for the LLM baselines against our transformer (beam width = 30). LLM cells report the mean with a 95% confidence interval over 100 problems per regime. The classical algorithm (RREFComPoly) solves none and is omitted. The best entry in each row is bolded.

| $(v_O, v_S)$ | Claude Haiku 4.5 | Gemini 2.5 Flash | GPT-4.1 mini | GPT-4o mini | DeepSeek v3.1 | Transformer (ours) |
|---|---|---|---|---|---|---|
| (2,3) | 0.28 [0.20, 0.38] | 0.06 [0.03, 0.13] | 0.04 [0.02, 0.10] | 0.03 [0.01, 0.09] | 0.00 [0.00, 0.04] | **0.69** |
| (3,2) | 0.06 [0.03, 0.13] | 0.09 [0.05, 0.16] | 0.03 [0.01, 0.09] | 0.01 [0.00, 0.05] | 0.00 [0.00, 0.04] | **0.15** |
| (3,3) | 0.23 [0.16, 0.32] | 0.05 [0.02, 0.11] | 0.02 [0.01, 0.07] | 0.01 [0.00, 0.05] | 0.00 [0.00, 0.04] | **0.24** |
| (3,4) | **0.30** [0.22, 0.40] | 0.03 [0.01, 0.09] | 0.01 [0.00, 0.05] | 0.01 [0.00, 0.05] | 0.00 [0.00, 0.04] | 0.28 |
| (4,3) | **0.20** [0.13, 0.29] | 0.05 [0.02, 0.11] | 0.08 [0.04, 0.15] | 0.02 [0.01, 0.07] | 0.00 [0.00, 0.04] | 0.09 |

**Classical algorithm.** We implemented and ran RREFComPoly (Guo et al., 2026), a recent reformulation of Faugère and Perret's MultiComPoly algorithm (Faugère & Perret, 2009a) and the classical line of work we compare against for multivariate decomposition. It assumes homogeneous inputs with as many inner polynomials as variables. Our problems are not homogeneous, so none of the 500 instances meet this precondition, and after homogenization three of the five regimes still violate the inner-count constraint. We ran it on 100 problems from each regime, and it solved none of the 500.

**Frontier LLMs.** We ran a zero-shot sweep of five non-reasoning models (Claude Haiku 4.5, Gemini 2.5 Flash, GPT-4.1 mini, GPT-4o mini, and DeepSeek v3.1) at 100 problems per regime. We use single-pass non-reasoning variants because they are the closest analogue to our model's single beam pass; reasoning-tier and tool-augmented multi-turn setups use orders of magnitude more compute and are excluded. Claude Haiku 4.5 is the strongest of the five, reaching 0.30 on $(3, 4)$, while the other four models stay at or below 0.09 on every regime (Table 5). Our transformer has the highest mean solve rate on three of the five regimes and falls within the best LLM's confidence interval on $(3, 4)$, leaving $(4, 3)$ as the only regime where it clearly trails. Its margin is largest on $(2, 3)$, where it reaches 0.69 against the strongest model's 0.28. Our model has orders of magnitude fewer parameters than the frontier LLMs and uses a single beam pass, yet it reaches comparable or better accuracy on most regimes. The classical algorithm is a less direct comparison as it assumes homogeneous inputs with as many inner polynomials as variables, a condition most of our instances do not satisfy, and on the problems we could run it returned no correct decompositions. This suggests that a compact model with a targeted RL finetuning pipeline can match much larger general-purpose models on this task, in a setting where established symbolic methods do not directly apply.

## 5 Conclusion

Our investigation into transformers for polynomial decomposition shows how neural networks can infer hidden algebraic structures. We find that model performance depends asymmetrically on polynomial complexity parameters ($\mathcal{D}_1$). Inner polynomial degree plays a dominant role, while outer polynomial complexity has limited impact. Counterintuitively, increasing the number of inner variables improves accuracy by imposing structural constraints, whereas more outer variables create information bottlenecks.

From an architectural viewpoint ($\mathcal{D}_2$), we confirm that performance scales with model size. We observe that fewer but more expressive attention heads are especially effective for this task. In terms of distribution adaptation ($\mathcal{D}_3$), models transfer rapidly to new coefficient distributions, requiring as little as 2% of the original training data, indicating that they internalize generalizable principles rather than rely on memorization. Moreover, we can enhance this generalization capability through strategic dataset design.

Beam search analysis ($\mathcal{D}_4$) yields up to $6.3\times$ improvement over greedy decoding for the SFT models, due to the sparse, precise nature of mathematical solutions. After fine-tuning with rank-aware BGRPO, beam-search accuracy increases by 33–37 percentage points over the SFT initialization and by 1.7–3.7 points over vanilla GRPO with non-overlapping $\pm1\sigma$ bands at every scale; greedy decoding alone after BGRPO already exceeds the SFT model's best beam-search score by 23.8–25.8 points at every scale. Finally, our model is competitive with non-reasoning, mid-size LLMs on most decomposition regimes with far fewer parameters and a single beam pass, a setting where the only available classical algorithm does not directly apply, and it also shows competitive performance in polynomial simplification against Mathematica's FullSimplify.

Our work provides, for the first time, a systematic analysis of transformer capabilities for polynomial decomposition through controlled experiments across four dimensions. Our methodologies can serve as a road map for exploring neural models in other domains that require non-local latent pattern discovery, such as functional decomposition problems ranging from systems engineering and mechanical design to digital logic design. While we developed BGRPO specifically for the polynomial decomposition problem, similar techniques may prove useful in other domains with sparse solution spaces where models can identify correct structures but struggle with specific details. We release the full pipeline, including data generation, pretraining, BGRPO, inference, evaluation, and baselines, on GitHub.

**Limitations** Our generalizability investigation was constrained to univariate polynomials with relatively narrow coefficient ranges and limited maximum degrees. Computational constraints restricted our architecture scaling experiments to relatively small models (maximum 6 layers, 768-dimensional embeddings); however, the consistent performance improvements without accuracy saturation suggest that further scaling would yield additional gains. Finally, our three rank-aware models converge to within 1 point of each other at the end of training (53.5% / 54.5% / 53.7% at $d = 256$ / $d = 512$ / $d = 768$), suggesting that the 200-problem multi-variable test set may impose a ceiling near 55% that limits our ability to distinguish the larger models from the smaller one; a more challenging evaluation set may show wider separation.

## Broader Impact Statement

This work investigates transformers applied to the polynomial decomposition problem, which is foundational research not tied to immediate societal applications. The synthetic data used in our experiments carries no privacy concerns, and the models are trained for a well-defined mathematical task. We note one indirect connection. Polynomial decomposition has applications in the cryptanalysis of certain multivariate public-key cryptosystems (Patarin & Goubin, 1997; Ye et al., 1999), so progress on decomposition could in principle inform attacks on such schemes. The cryptosystems in question are already known to be vulnerable and of largely historical interest, and our models operate far below the scale needed for any practical cryptanalytic use, so we see no direct path to harmful applications.

## Acknowledgments

We would like to express our gratitude to Aike Liu for initiating this project with NS and for her valuable contributions during the early stages. Her initial insights and discussions were instrumental in shaping this work. We thank David Simmons-Duffin for granting JL and NS access to computational resources that were essential for exploring and assessing the potential of this project. A significant portion of our experiments was conducted using the High-Performance Computing facilities at Caltech, supported by a grant to TYY. JL and NS's work at Caltech was supported in part by Simons Foundation grant 488657 (Simons Collaboration on the Nonperturbative Bootstrap).

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

## A   Backward Synthetic Data Generation Algorithm

Our backward synthetic data generation in subsection 2.1 can be described as follows.

---
**Algorithm 1** Backward Generation of Synthetic Training Data

---
**Require:** Coefficient range $C_{\text{inner}}$, $C_{\text{outer}}$; maximal degrees $d_{\text{inner}}$, $d_{\text{outer}}$; variable counts $v_{\text{inner}}$, $v_{\text{outer}}$; term limits $t_{\text{inner}}$, $t_{\text{outer}}$.
1: Generate outer polynomial $g$ with $v_{\text{outer}}$ variables, coefficients $\in C_{\text{outer}}$, degree $= d_{\text{outer}}$, and no more than $t_{\text{outer}}$ monomial terms.
2: Generate $v_{\text{outer}}$ inner polynomials $h_1, \ldots, h_{v_{\text{outer}}}$, where each $h_i$ has $v_{\text{inner}}$ variables, coefficients $\in C_{\text{inner}}$, degree $= d_{\text{inner}}$, and no more than $t_{\text{inner}}$ monomial terms.
3: $f \leftarrow g(h_1, \ldots, h_{v_{\text{outer}}})$, i.e. substitute $h_1, \ldots, h_{v_{\text{outer}}}$ into $g$, expand and collect the monomial terms.
4: **return** $(f,\ g,\ h_1, \ldots, h_{v_{\text{outer}}})$

---

## B   Tokenization

We encode polynomials using prefix notation, with separate tokens for operators, digits, and variables. Each number includes its sign, so we only use addition, multiplication, and power operators. Subtraction is represented as addition with a negative sign. We experimented with several tokenization schemes for the digits, such as grouping multiple digits into a single token (for example two- or three-digit tokens) and inserting an auxiliary separator token between digits (for example representing 123 as $1 \sim 2 \sim 3$). These tricks helped slightly when training data was scarce, but with a large amount of training data the difference became minimal, so we adopted the simplest scheme. Each input sequence consists of the tokenized expanded polynomial $f$ followed by a question mark token '?'. The target output format depends on the number of outer variables. For $v_{\text{outer}} = 1$, the target output is simply the tokenized inner polynomial $h$; for $v_{\text{outer}} > 1$, the target output begins with the tokenized outer polynomial $g$ followed by each tokenized inner polynomial $h_1, \ldots, h_{v_{\text{outer}}}$, with all polynomials separated by a delimiter token '&'.

Below is an example of a tokenized training input 'x' and target output 'y':

$x : + * \text{P } 9\ 0\ a + * \text{N } 3\ 1\ 9\ \hat{}\ a\ \text{P } 2 + * \text{N } 3\ 6\ \hat{}\ a\ \text{P } 3 * \text{N } 1\ \hat{}\ a\ \text{P } 4\ ? + \text{N } 5 + * \text{P } 1\ 8\ a\ \hat{}\ a\ \text{P } 2\ \square \ldots$
$y : \square\square\square\square\square\square\square\square\square\square\square\square\square\square\square\square\square\square\square\square\square\square\square\square\square\square\square\square\square\square\square\square\square\square\square + \text{N } 5 + * \text{P } 1\ 8\ a\ \hat{}\ a\ \text{P } 2\ \square\square \ldots$

This example shows a training pair where the outer polynomial is $90a - 319a^2 - 36a^3 - a^4$ and the target inner polynomial is $-5 + 18a + a^2$. The $\square$ symbol represents a padding token which is excluded from the log-likelihood loss calculation.

## C   Example Output Logits and Effectiveness of the Beam Search

Figure 10 shows example top-3 probabilities for each token position in the answer sequence at temperature 1, using the layer-6, embedding dimension 512 model from our $\mathcal{D}_2$ experiments. Correct answers are highlighted in red. The visualization clearly illustrates that the model's primary source of confusion occurs in sign decisions, while it confidently predicts most of the other token types.

Table 6 quantifies this observation by showing the probability and accuracy statistics for different token types across our model architectures from $\mathcal{D}_2$. These statistics were computed using a test set of 1000 polynomial decomposition problems at temperature 1.

As discussed in Section 2.2, our models achieve approximately 90% accuracy when predicting non-sign tokens, but exhibit near-random performance when choosing between positive and negative signs. This specific error pattern makes beam search particularly effective for our task.

The effectiveness of beam search stems from its ability to explore multiple sign configurations while preserving the high-confidence structural tokens. In probability terms, selecting a token with 0.1 probability instead of one with 0.9 probability is equivalent to making approximately 11 consecutive choices of a 0.45 probability

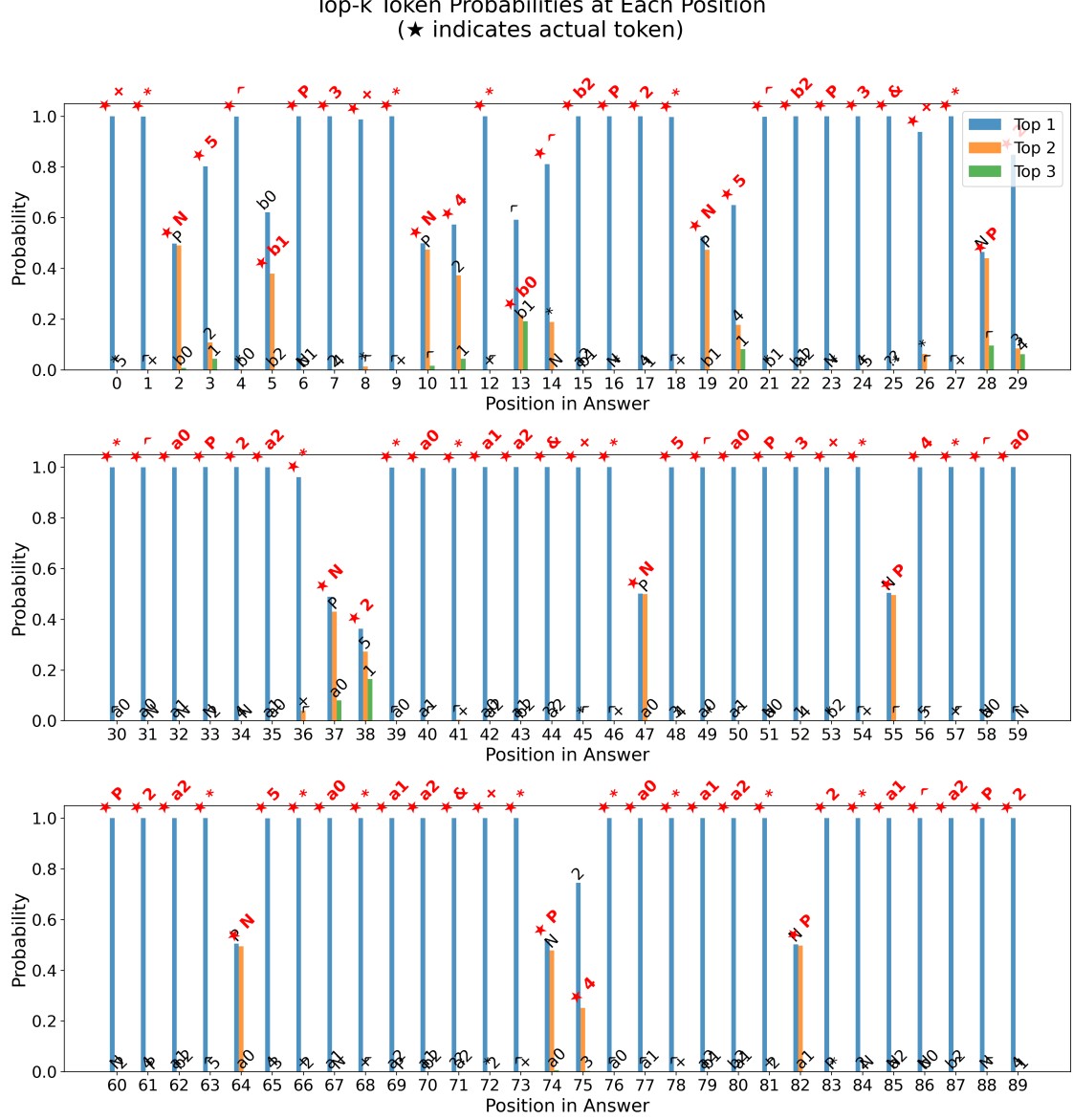

Figure 10: Top-3 probability for each token position in the answer sequence where

**Answer:** + * N 5 ^ b1 P 3 + * N 4 * b0 ^ b2 P 2 * N 5 ^ b2 P 3 & + * P 2 * ^ a0 P 2 a2 * N 2 * a0 * a1 a2 & + * N 5 ^ a0 P 3 + * P 4 * ^ a0 P 2 a2 * N 5 * a0 * a1 a2 & + * P 4 * a0 * a1 a2 * P 2 * a1 ^ a2 P 2

**Question:** + * P 6 2 5 ^ a0 P 9 + * N 1 5 0 0 * ^ a0 P 8 a2 + * P 1 8 7 5 * ^ a0 P 7 * a1 a2 + * P 1 2 0 0 * ^ a0 P 7 ^ a2 P 2 + * N 3 0 0 0 * ^ a0 P 6 * a1 ^ a2 P 2 + * P 1 8 7 5 * ^ a0 P 5 * ^ a1 P 2 ^ a2 P 2 + * N 3 2 0 * ^ a0 P 6 ^ a2 P 3 + * P 1 2 0 0 * ^ a0 P 5 * a1 ^ a2 P 3 + * N 1 6 2 8 * ^ a0 P 4 * ^ a1 P 2 ^ a2 P 3 + * P 4 3 3 * ^ a0 P 3 * ^ a1 P 3 ^ a2 P 3 + * N 1 2 8 * ^ a0 P 3 * ^ a1 P 2 ^ a2 P 4 + * N 3 5 2 * ^ a0 P 2 * ^ a1 P 3 ^ a2 P 4 + * N 3 2 * ^ a0 P 2 * ^ a1 P 2 ^ a2 P 5 + * N 2 0 8 * a0 * ^ a1 P 3 ^ a2 P 5 * N 4 0 * ^ a1 P 3 ^ a2 P 6 ?

Table 6: Token Type Analysis Across Different Model Architectures

| Token Type | Metric | 4 Layers | | | 6 Layers | | |
| --- | --- | --- | --- | --- | --- | --- | --- |
| | | 256 dim | 512 dim | 768 dim | 256 dim | 512 dim | 768 dim |
| Sign | Probability | $0.489 \pm 0.001$ | $0.489 \pm 0.001$ | $0.493 \pm 0.001$ | $0.491 \pm 0.001$ | $0.490 \pm 0.001$ | $0.490 \pm 0.001$ |
| | Accuracy | $0.519 \pm 0.006$ | $0.531 \pm 0.006$ | $0.530 \pm 0.006$ | $0.522 \pm 0.006$ | $0.523 \pm 0.006$ | $0.521 \pm 0.006$ |
| Operator | Probability | $0.920 \pm 0.002$ | $0.915 \pm 0.002$ | $0.919 \pm 0.002$ | $0.927 \pm 0.002$ | $0.925 \pm 0.002$ | $0.925 \pm 0.002$ |
| | Accuracy | $0.937 \pm 0.002$ | $0.934 \pm 0.002$ | $0.935 \pm 0.002$ | $0.943 \pm 0.002$ | $0.941 \pm 0.002$ | $0.942 \pm 0.002$ |
| Number | Probability | $0.880 \pm 0.002$ | $0.870 \pm 0.002$ | $0.878 \pm 0.002$ | $0.890 \pm 0.002$ | $0.885 \pm 0.002$ | $0.884 \pm 0.002$ |
| | Accuracy | $0.901 \pm 0.002$ | $0.893 \pm 0.003$ | $0.897 \pm 0.002$ | $0.911 \pm 0.002$ | $0.905 \pm 0.002$ | $0.903 \pm 0.002$ |

Note that values shown as mean $\pm$ standard error of the mean. The sign token probabilities are near-random, while operators and numbers show high confidence and accuracy.

token over a 0.55 probability token. Since our polynomial expressions typically contain fewer than 10 sign decisions, beam search with a width of approximately 30 can efficiently cover most viable sign permutations while maintaining the correct monomial structure identified with high confidence.

## D  Attention Score Analysis: Monomial Heads

Attention mechanism analysis has provided valuable insights into transformer model behaviors, with studies identifying specialized attention heads that serve specific functions. For example, Olsson et al. (2022) identified "Induction Heads" that play a central role in in-context learning, while Wang et al. (2022) provided a detailed analysis of indirect object identification in GPT-2 Small.

In our analysis of attention patterns in polynomial decomposition models, we identified specialized attention heads that recognize the structure of polynomials, particularly focusing on monomial identification. We call these "Monomial Heads," and they appear consistently across all model sizes in our architecture scaling experiments ($\mathcal{D}_2$).

Monomial Heads manifest in two distinct patterns in our models. First, in layer 0, several attention heads consistently attend to tokens 1-5 positions behind the current position, as shown in the leftmost plot of Figure 11. Second, in layer 1, we observe specialized behavior where certain heads focus attention on specific tokens within each monomial of the input polynomial (middle plot), while others specifically attend to delimiter tokens in the decomposition output (rightmost plot).

We hypothesize that this represents a two-stage process. In layer 0, the model identifies tokens that serve as monomial indicators by examining local context (1-5 tokens behind). In layer 1, tokens within each monomial attend to these indicator tokens to establish their monomial membership. While this pattern is most clear in the encoding of the input polynomial, the decomposition output shows evidence of boundary recognition, particularly at the transitions between inner functions marked by delimiter tokens.

## E  Correctness Verification

We judge a predicted decomposition correct by recomposition rather than by matching a fixed ground-truth string. Given a predicted outer polynomial and predicted inner polynomials, we substitute the inner polynomials into the outer one, expand the result, and compare it to the original target under exact symbolic equality. A prediction is counted correct if and only if this recomposition reproduces the target exactly.

This handles the non-uniqueness of decompositions. A target may admit several distinct valid decompositions, and any prediction that recomposes to the target is accepted, even when it differs from the decomposition used to generate the problem. We use the same check in all three settings where correctness is assessed: as the reward signal during BGRPO training, during inference and evaluation, and when scoring the classical and LLM baselines. Using a single verifier throughout keeps the comparison across methods consistent.

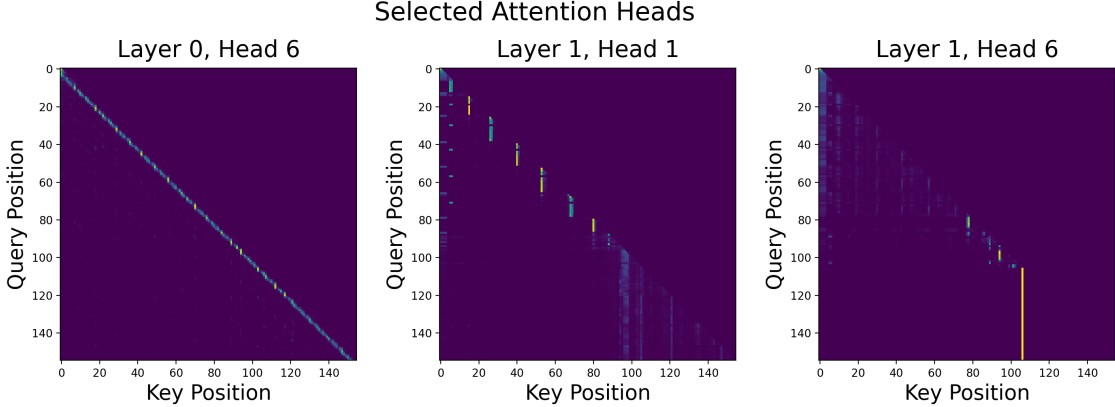

Figure 11: Attention score visualization of selected attention heads from our 6-layer transformer model with embedding dimension 768. The visualization shows attention patterns for a tokenized polynomial sequence and its decomposition.

**Input polynomial:** + * P 2 5 6 ^ a0 P 9 + * N 1 9 2 * ^ a0 P 8 a1 + * P 4 8 * ^ a0 P 7 ^ a1 P 2 + * N 4 * ^ a0 P 6 ^ a1 P 3 + * N 6 4 * ^ a0 P 3 ^ a1 P 6 + * P 1 6 * ^ a0 P 2 ^ a1 P 7 * P 6 4 ^ a1 P 9 ?

**Model's decomposition output:** + * N 4 ^ b0 P 3 + * b0 ^ b2 P 2 * N 1 ^ b2 P 3 & + * N 4 ^ a0 P 3 * ^ a0 P 2 a1 & + * N 3 ^ a1 P 3 + * N 2 * a1 ^ a2 P 2 * N 4 ^ a2 P 3 & * N 4 ^ a1 P 3

The visualization reveals how different attention heads focus on specific structural elements when decomposing polynomials.

# F  Hyperparameter Justification

We bound every hyperparameter so that the task tests structural identification rather than large-integer arithmetic. Decomposition depends on which monomials appear, how variables group, and how signs attach. With bounded coefficients, errors concentrate on sign tokens rather than on arithmetic tokens (Table 6). The full per-axis configuration is in Table 1.

Each axis varies one factor and fixes the rest to isolate a single source of difficulty. We sweep degree in $\mathcal{D}_1$ (degree scaling) and variable structure in $\mathcal{D}_1$ (variable scaling), so both are fixed at 3 in $\mathcal{D}_2$ to isolate architecture effects. We use the tightest coefficient range, $[-5, 5]$, for the multivariate structural experiments ($\mathcal{D}_1$ variable scaling and $\mathcal{D}_2$). The degree-scaling and adaptation axes use wider ranges suited to their focus, and we keep adaptation univariate to separate distribution shift from variable structure.

The five multivariate regimes $(v_{\text{outer}}, v_{\text{inner}}) \in \{(2,3), (3,2), (3,3), (3,4), (4,3)\}$ vary difficulty along two dimensions, the number of outer variables $v_{\text{outer}}$ (equivalently, the number of inner polynomials) and the variables per inner polynomial $v_{\text{inner}}$. Because we sample the outer freely within the degree and term caps, a larger-regime sample whose outer uses only a subset of its variables effectively becomes a smaller-regime instance, so each regime spans a range of effective difficulties.

The space of valid decompositions remains combinatorially large even with bounded coefficients, and external baselines confirm the difficulty. The classical algorithm returns 0/500 across the five regimes, and the strongest non-reasoning, mid-size LLM (Claude Haiku 4.5) reaches only 0.30 on $(3, 4)$, while the other four models stay at or below 0.09 on every regime (Table 5).

# G  Dataset Construction and Splits

For each evaluation axis we generate train, validation, and test sets independently. Training sets contain 1M to 2M instances depending on the axis (see Section 3.2 for per-axis details), with 128 validation and 700 test instances per regime. We filter test instances against training and found no overlaps.

We use held-out evaluation sets of differing sizes across experiments. For the frontier-LLM and classical-algorithm baselines (Section 4.7) and the simplification comparison (Section 4.6) we use 100 problems per regime (500 total). For the BGRPO experiments we report on 200 held-out multi-variable problems (Section 4.5).

For the simplification comparison we evaluate the 6-layer, 512-dim model from the $\mathcal{D}_2$ sweep after rank-aware BGRPO, on the same 500-instance held-out test set and overlap filter. Generation settings follow Section 3.2.

## H  Inference Efficiency

Standard beam search runs one forward pass per beam per decode step, so inference FLOPs scale linearly in beam width, not quadratically as we assumed in an earlier version. The strongest efficiency result follows directly from Table 3. After rank-aware BGRPO, greedy decoding (beam width 1) already exceeds the SFT model's beam@30 accuracy at every scale by 23.8 to 25.8 percentage points. This eliminates beam-search overhead entirely at matched or better accuracy, replacing 30 forward passes per decode step with one. With beam search on top (beam@30), our post-BGRPO model reaches 53.5–54.5%, well beyond SFT beam@30 at 16.5–20.0%. BGRPO therefore offers two deployment modes. We can use greedy decoding with no beam overhead that already surpasses SFT beam@30, or we can apply beam search for roughly 9–11 additional accuracy points when compute allows.

