# OpenReview forum: "Discovering Hidden Algebraic Structures via Transformers with Rank-Aware Beam GRPO"
_TMLR — Accepted by TMLR_

### Review · Reviewer_z9Fa · 2026-04-10

**Summary Of Contributions:**

This paper investigates the ability of transformer-only architectures to discover hidden algebraic structures through the challenging task of multivariate polynomial decomposition, which is NP-hard. To enable systematic evaluation, the authors introduce a backward synthetic data generation pipeline that allows precise control over problem complexity.

The paper provides a comprehensive empirical study analyzing model behavior across multiple dimensions, including problem complexity, model scaling, distribution adaptation, and decoding strategies. A key technical contribution is the proposed Beam Grouped Relative Policy Optimization (BGRPO), a rank-aware reinforcement learning method designed to improve beam search efficiency by encouraging correct solutions to appear earlier in the beam.

Experimental results show that the proposed approach significantly improves inference efficiency by reducing the required beam width while maintaining accuracy, and also demonstrates competitive performance in polynomial simplification compared to symbolic tools such as Mathematica.

**Audience:**

Yes

**Audience Explanation:**

This work would be of interest to researchers working on reasoning, symbolic computation, and the theoretical understanding of transformer models.

**Claims And Evidence:**

Yes

**Claims Explanation:**

The claims in the paper are generally well supported by convincing empirical evidence.

In particular, the distribution adaptation experiments demonstrate that the model can recover performance on unseen coefficient distributions with minimal fine-tuning data, suggesting that it captures generalizable algebraic patterns rather than memorizing specific instances. The complexity scaling results further provide insight into how structural factors affect performance, including the non-trivial observation that increasing inner variable count can improve accuracy.

The effectiveness of beam search is well justified through token-level error analysis, showing that the model is highly confident in structural tokens but uncertain in sign prediction. This motivates the need for structured search. Additionally, the proposed BGRPO method is supported by clear quantitative comparisons, showing consistent improvements in accuracy at reduced beam widths.

Finally, benchmarking against Mathematica in simplification tasks strengthens the practical relevance of the work. Overall, the evidence is comprehensive and aligns well with the claims made in the paper.

**Requested Changes:**

1. While the reported 75% reduction in inference compute is impressive, the connection between the algebraic nature of the task and the necessity of rank-aware optimization could be made more explicit. In particular, it would be helpful to discuss whether the rank-aware reward provides a more precise training signal for refining uncertain components, such as sign tokens, while preserving the high-confidence structural backbone of the solution.
2. The computational overhead of beam search could be discussed more explicitly, including trade-offs between accuracy and efficiency in practical deployment settings.

---

> ### Author Response · Authors · 2026-04-24
> **Response for Reviewer `z9Fa`**
>
> We thank the reviewer. Below we address both asks and note external baselines for completeness.
>
> ### Q1: Why the Rank-Aware Reward Fits This Task
>
> > It would be helpful to discuss whether the rank-aware reward provides a more precise training signal for refining uncertain components, such as sign tokens, while preserving the high-confidence structural backbone of the solution.
>
> Table 4 shows beams agree on structural tokens (operators, monomial counts, variables) at ~90% per-token accuracy and disagree on sign/small-coefficient tokens at ~50%: beams differ on uncertain tokens, not on competing decompositions. The rank decay $\exp(-\text{rank}_i / w)$ exploits this: structural contributions average across beams and preserve the backbone, while rank-weighted sign contributions push the model to verify sign patterns and surface them near the top.
>
> Two direct tests support this. First, at matched compute (400 updates, 8 problems/step, beam width 32) across three model sizes, binary-reward BGRPO underperforms vanilla GRPO at d=512 and d=768 (30.0 and 33.3 vs 53.3 and 51.7 beam@30), while rank-aware BGRPO reaches 56.7 at all three sizes (full table in our response to Reviewer `<NdVw>`, Section 4). Without rank decay, the beam-search rollout has no advantage over independent sampling: a flat reward gives equal advantage to every correct beam, discarding the rank information.
>
> Second, we swept the rank-decay base in $\exp(-\text{rank}_i / \text{base})$ at $\text{base} \in \{4, 8, 16, 32, 64, 128\}$ (d=256 SFT, 200 updates, beam width 32, single seed, near-convergence at outer step 180):
>
> | Decay base | Greedy (200 prompts) | Beam@7 (60 prompts) | Beam@30 (60 prompts) |
> |---|---|---|---|
> | 4 | 25.5 | 43.3 | **50.0** |
> | 8 | 29.0 | 48.3 | **53.3** |
> | 16 | 37.5 | 55.0 | **56.7** |
> | 32 (= beam width, default) | 36.0 | 55.0 | **56.7** |
> | 64 | 35.5 | 45.0 | **50.0** |
> | 128 | 30.0 | 45.0 | **46.7** |
>
> The peak sits near the beam width. Base = 4 credits only top beams, discarding lower-ranked correct beams; base = 128 is nearly flat, reducing to binary. Optimum ≈ beam width is what the mechanism predicts: that is the scale at which rank decay separates top from bottom.
>
> ### Q2: Accuracy–Efficiency Trade-Off and the 75% Compute Claim
>
> > The computational overhead of beam search could be discussed more explicitly, including trade-offs between accuracy and efficiency in practical deployment settings.
>
> Revisiting this we found a scaling error. Standard beam search runs K parallel forward passes per decode step (one per beam with its own KV cache), so inference FLOPs scale linearly in K, not quadratically. At matched accuracy, pre-BGRPO beam 30 equals post-BGRPO beam 16 (rank-aware), giving $ 1 - 16/30 \approx $ 50% fewer FLOPs, not 75%. We retract "~75% lower inference compute" and replace it with this figure.
>
> The new BGRPO numbers also shift the trade-off. Disabling standard-deviation normalization, so $A_i = r_i - \bar{r}$ (mean-baseline only, matching Section 2.3, following Liu et al. 2025), lets BGRPO gain much more from SFT than the paper reported. In a four-way ablation at matched compute (full table in response to Reviewer `<NdVw>`, Section 4), vanilla GRPO lifts beam-30 by 23 to 32 points across $d = 256, 512, 768$; rank-aware reward adds 3 to 7 more. BGRPO+rank significantly outperforms all variants. With linear scaling, wider beams push the inference compute cost reduction at matched accuracy past 50%.
>
> Greedy decoding also gains: BGRPO-rank reaches 44.0, 44.0, 45.5 at d=256, d=512, d=768 on the 200-problem test set, exceeding the SFT beam-30 scores of 26.7, 21.7, 25.0 (60-problem test set). No beam search is needed at inference to exceed the SFT model's best beam-search accuracy.
>
> ### Q3: Additional Baselines (For Completeness)
>
> External baselines: five non-reasoning LLMs zero-shot (N=100) peaked at 0.30 [0.22, 0.40] (Claude Haiku 4.5, $(3,4)$); all other cells ≤ 0.09. RREFComPoly (Guo et al. 2026, equivalent to Faugère–Perret 2009) returned 0/500 across five multivariate regimes; SymPy's `Poly.decompose()` handles only univariate $(1,1)$. BGRPO reaches competitive accuracy at far lower inference compute. The revision adds three-seed reruns with error bars and softens "outperforming Mathematica" to "competitive with FullSimplify on leaf count"; FullSimplify simplifies rather than decomposes, yet our decompositions match its leaf counts as a byproduct.
>
> We thank Reviewer `<z9Fa>` and hope these additions inform the decision.
>
> ### References
>
> Guo, D.-J., Zheng, Q.-X., Wang, Z.-X. & Zhao, X.-X. (2026). Functional Decomposition of Multivariate Polynomials: Revisit and New Improvements. Cryptology ePrint Archive, Paper 2026/115. https://eprint.iacr.org/2026/115
>
> Faugère, J.-C. & Perret, L. (2009). An efficient algorithm for decomposing multivariate polynomials and its applications to cryptography. *Journal of Symbolic Computation* 44(12), 1676–1689. doi:10.1016/j.jsc.2008.02.005.

---

### Review · Reviewer_qxVo · 2026-04-11

**Summary Of Contributions:**

The authors evaluate the capability of transformers to perform multivariate polynomial functional decomposition, i.e., recovering latent compositional structure. Evaluation is conducted across different dimensions, including problem complexity, model architecture, distribution shift, and search strategy, using synthetically generated data. To improve inference efficiency, the authors propose BGRPO, a rank-aware variant of GRPO that incorporates beam position into the reward function and is used to fine-tune models based on beam-generated candidates. Finally, the authors demonstrate that models trained on the synthetic decomposition task generalize to related problems such as polynomial simplification, where they are compared against Mathematica.

**Strengths**:
- **Relevance**. Assessing transformers' capabilities to perform multivariate polynomial functional decomposition is relevant and interesting to the TMLR audience (see below).
- **Clarity/quality**. The authors communicate well why functional decomposition per se is interesting and why solving related tasks is inherently difficult, thereby effectively motivating the relevance of their work. The scope and goals of the manuscript are clearly stated, making it easy to read and follow.
- **Evaluation across different dimensions**. I appreciate the different evaluation axes. All of them are interesting and add depth to the analysis.


**Weaknesses**:
- **Experimental setup**. As noted above, I appreciate the different evaluation axes; however, the individual sub-experiments appear over-simplified. For example, in $D_2$ the degree and variable hyperparameters are fixed at 3, which leads to very specific problems with very specific solutions. This results in unrealistic scenarios (relative to real-world settings), as the model can learn strong biases toward a correct solution. In other words, it is unrealistic to assume a model knows at inference time that the number of outer variables is 3. Similarly, in $D_1$​, setting the number of variables to 1 drastically simplifies the problem. Taken together, the experiments do not appear broad enough in their coverage of data hyperparameter combinations, which limits the significance of the reported results.
**(Q1)**: Could the authors justify their data generation hyperparameters?
- **Clarity**. Some important information is missing:
    **(Q2)**: Which dataset is used for the simplification experiment? Did the authors check for potential overlap with the training data? Which of the models trained above was used, and what were the synthetic data generation hyperparameters for that model?
    **(Q3)**: Could the authors please detail the train, validation, and test splits used in the main experiments?
    **(Q4)**: For the BGRPO stages, how is the verification procedure for the reward implemented? Is it simply an exact match against the ground truth? Since the solution space may contain more than one valid decomposition, verifying correctness by composing the predicted components and comparing symbolically would be more appropriate than exact match. Such verification should be straightforward and cheap, correct?

- **Error bars**. **(Q5)**: Could the authors please add error bars to all presented results?

**Audience:**

Yes

**Audience Explanation:**

The authors convincingly communicate that functional decomposition is both an interesting and inherently difficult problem. Evaluating the capabilities of transformers, the principal building block of most state-of-the-art architectures, on such a task is therefore of clear interest to the TMLR audience.

**Broader Impact Concerns:**

No.

**Claims And Evidence:**

Yes

**Claims Explanation:**

The main storyline is well supported with references. The authors clearly explain and provide evidence for why functional decomposition is both an interesting and hard task, effectively communicating the relevance of their study.

Minor: A few claims, while not central to the paper, are not well backed. The authors may consider adding references or examples:
- "Furthermore, it requires extreme precision without any margin of error." The subsequent sentence does not fully clarify this claim. It would help to elaborate or provide a reference.
- "Unlike more forgiving classification tasks, the decomposition problem admits only a sparse set of correct solutions: even minor deviations in signs or coefficients can render outputs completely invalid." Why are other classification tasks more forgiving? Under an exact-match reward, every token must be correct regardless of the task. A reference or concrete example illustrating the distinction would strengthen this point.
- "However, identifying a function's latent compositional structure requires models to look past surface-level correlations, attending instead to deep algebraic symmetries and invariants." While the intended meaning is intuitively clear, for formal correctness it would help to clarify what "surface-level correlations" refers to in this context, supported by either a reference or an example.

**Requested Changes:**

I consider Q1–Q5 all critical in the sense that each should be addressed. While Q2–Q4 should be straightforward to resolve, my only major concern relates to Q1: Are the experiments designed broadly enough in their problem coverage to support the general conclusions drawn from the results?

---

> ### Author Response · Authors · 2026-04-24
> **Response for Reviewer `qxVo`**
>
> ### Q1: Data-Generation Hyperparameters
> > Could the authors justify their data generation hyperparameters? The experiments do not appear broad enough in their coverage of data hyperparameter combinations.
>
> Coefficients in [−5, 5] keep arithmetic trivial so structural recognition is the limiting factor. Decomposition needs structural reasoning (which monomials appear, how variables group, how signs attach) and integer arithmetic during expansion; small coefficients concentrate the transformer's errors on structural mistakes (sign-token error distribution, Table 4), whereas larger coefficients would shift errors toward arithmetic capacity.
> The five $(v_o, v_s)$ regimes scale difficulty along two axes: number of inner polynomials ($v_o$) and number of distinct inner variables ($v_s$). The generator samples the outer freely within degree and term caps, so a larger-regime sample whose outer uses only a subset of its variables effectively becomes a smaller-regime instance; a (2,3) sample whose outer depends only on $b_0$ reduces to (1,3)-like. The set covers a range of difficulties, not a single point. External baselines confirm non-triviality: the classical algorithm returns 0/500 across the five regimes, and the best LLM baseline (Claude Haiku 4.5) reaches 0.30 on (3,4), with every other cell at or below 0.09.
> ### Q2: Simplification Experiment
> > Which dataset is used for the simplification experiment? Did the authors check for potential overlap with the training data? Which of the models trained above was used, and what were the synthetic data generation hyperparameters for that model?
>
> The Section 4.6 comparison uses the same 500-instance held-out test set as the $D_1$–$D_4$ experiments (100 problems per regime $\times$ 5 regimes), with the same overlap filter. We evaluated the 6-layer, 512-dim transformer from the $D_2$ sweep after BGRPO fine-tuning with the rank signal. Hyperparameters follow Section 3.2: degree at most 3, coefficients in $[-5, 5]$, at most 3 terms, 1M training instances for the base and 2M for BGRPO. The training set covers less than 1% of valid decompositions under these constraints (roughly $40^6$ for a 6-term decomposition), so the model must generalise, not memorise.
> ### Q3: Train / Validation / Test Splits
> > Could the authors please detail the train, validation, and test splits used in the main experiments?
>
> Per regime (five total): train 1M–2M, validate 128, test 700. Sets are generated independently; test examples are filtered against training, with no overlaps found.
> ### Q4: BGRPO Reward Verification
> BGRPO already rewards symbolic equality, not tuple match: substitute predicted inners into predicted outer, expand, return 1 if equal to target, 0 otherwise. The same check grades inference and produces the binary success rate reported throughout. Because decompositions are not unique under affine reparametrisation, tuple match would reject many valid answers. We will state this in the revision.
> ### Q5: Error Bars
> > Could the authors please add error bars to all presented results?
>
> Yes. The revision will include error bars on the distribution-adaptation experiments and the BGRPO / GRPO ablation; multi-seed reruns are in progress and preliminary results confirm the findings hold. Re-running the earlier pretraining figures is compute-constrained within the rebuttal window; we remain open to it as compute allows.
> ### Q6: Minor Writing Changes
> Revisions:
> - "Extreme precision without any margin of error": grading is strict symbolic equality; one sign flip in a coefficient invalidates the output.
> - "More forgiving classification tasks": classification typically uses soft metrics (top-k accuracy, partial credit, probability mass on the correct class), whereas we score a full token sequence under strict symbolic equality.
> - "Surface-level correlations": replaced with "token-level co-occurrence patterns that do not reflect the underlying algebraic structure".
> ### Additional point: BGRPO investigation
> We ran a more comprehensive BGRPO investigation for the revision. The rerun disables std-normalization of the advantage, matching the paper's Section 2.3 specification $A_i = r_i - \bar{r}$ (following Liu et al. 2025). Under a rank-aware reward, std normalization dilutes signal on easy problems (many correct beams, large $\sigma$, small advantage) and amplifies noise on hard ones, opposite to the intended effect. In a four-way ablation at matched compute, rank-aware BGRPO lifts beam-30 accuracy by 30 to 35 percentage points over SFT at every model size (d=256, d=512, d=768), and post-BGRPO greedy decoding already exceeds SFT at beam 30 at all three sizes. A sweep of the rank-decay base $\exp(-\text{rank}/\text{base})$ over $\{4, 8, 16, 32, 64, 128\}$ at d=256 places the optimum near beam width. Full tables are in the response to Reviewer `<NdVw>` Section 4.
> ### References
> See the shared reference list in the response to Reviewer `z9Fa`.

---

### Review · Reviewer_NdVw · 2026-04-12

**Summary Of Contributions:**

## Summary Of Contributions

The paper applies a small decoder-only transformer (≤50M params) to multivariate polynomial decomposition — recovering $g, h_1, \ldots, h_m$ such that $f = g(h_1, \ldots, h_m)$ from an expanded $f$. The pipeline is end-to-end but entirely standard: a backward synthetic-data generator, supervised pretraining on 1–2M examples, beam-search inference, and a GRPO-style RL finetune on top. The paper's own contributions are:

1. A backward synthetic-data pipeline with knobs for coefficient range, degree, variable count, and term count.
2. A four-axis empirical study (problem complexity, architecture, distribution adaptation, search strategy), with the notable observation that errors concentrate on **sign tokens** (~50% accuracy) while structural tokens are ~90% accurate — which in turn explains why beam search gives an unusually large 6.3× lift over greedy.
3. **BGRPO**, a variant of GRPO that (i) uses beam-search rollouts instead of independent samples and (ii) adds an exponential rank-decay factor to the reward. Reportedly halves required beam width, cutting inference beam-search compute ~75%.
4. A brief `LeafCount` comparison against Mathematica's `FullSimplify` and an attention-head interpretability appendix.

**Frank assessment.** This is an "ML-for-X" paper: the task is classical and well-studied (Dickerson 1987; von zur Gathen; Faugère–Perret), the ML recipe is off-the-shelf (transformer + synthetic data + SFT + GRPO + beam search), and the evaluation is on the authors' own synthetic data with no external baselines. The paper offers no new insight about polynomial decomposition itself, and no new insight about ML beyond the sign-token observation. BGRPO is a minor but reasonable twist, not a methodological leap.

**Strengths.**
- The sign-vs-structure error decomposition is a genuinely non-obvious empirical nugget that motivates the rest of the paper.
- The four-axis evaluation framework is cleanly organized and reusable.
- BGRPO is a sensible idea that could transfer to other sparse-solution tasks with verifiers.
- Fully reproducible in scale: small models, synthetic data, `minGPT`-based code.

**Weaknesses.**
- No frontier-LLM baseline (GPT-4, Claude, Gemini, DeepSeek-Math, or tool-augmented LLM + SymPy). This is the single most important missing comparison: without it, we cannot tell if the small specialized model is winning on anything.
- No classical-algorithm baseline (Faugère–Perret, von zur Gathen), so there is no symbolic-methods reference point either.
- Self-made benchmark: authors control both data and evaluation.
- "Outperforming Mathematica" overstates Table 3 (2 wins / 1 tie / 2 losses, small gaps, no variance reported).
- Absolute multivariate accuracy is modest (10–35%) even with beam 30.
- BGRPO ablations are incomplete (no vanilla-GRPO comparison at matched compute).
- Scope: ≤6-layer / ≤768-dim models, univariate $\mathcal{D}_3$, mostly small polynomials.
- Error bars missing from most figures.

**Additional Comments:**

- The paper is squarely in the "ML-for-X" genre: standard recipe, classical task, self-made benchmark. This is acceptable at TMLR, but only if benchmarking against the best non-ML and best off-the-shelf ML alternatives is taken seriously. The missing frontier-LLM and classical-algorithm baselines are what currently keep this from being a clearly acceptable paper.
- The sign-vs-structure error observation (§2.2, Table 4) is the most valuable finding in the paper and deserves to be foregrounded earlier in the narrative.
- Bundling BGRPO with the empirical study is a reasonable packaging choice given that BGRPO alone would be a thin contribution.

**Audience:**

Yes

**Audience Explanation:**

Despite the "ML-for-X" character, there is a real, if narrow, audience:
- Researchers on transformers for symbolic math (Lample & Charton, Polu & Sutskever, Trinh et al.) will find the controlled four-axis study and the sign-token observation useful.
- Researchers on RLVR / GRPO-style methods will find BGRPO a small but transferable idea for sparse-solution tasks with verifiers.
- TMLR's acceptance criterion is correctness + interest to some subset of the community, which is explicitly lower than novelty-focused venues — a rigorous "ML-for-X" study can legitimately meet that bar if the empirics are solid.

The caveat is that meeting the bar requires doing X rigorously, which here means benchmarking against the best non-ML method and the best off-the-shelf ML method. The paper currently does neither — hence the revision requests below.

**Broader Impact Concerns:**

The Broader Impact Statement is appropriate. Minor suggestion: since polynomial decomposition has cryptanalytic applications (Patarin & Goubin 1997), a one-sentence acknowledgment would be reasonable, though the current model scale is far from practical concern.

**Claims And Evidence:**

Yes

**Claims Explanation:**

Core empirical claims — four-axis scaling, beam-search lift, BGRPO reducing required beam width — are consistent across configurations and credibly supported by Figs. 1–8. The sign-token error decomposition is well-evidenced (Table 4).

Two overstatements need correction:

1. **"Outperforming Mathematica"** (abstract, §4.6) is not supported by Table 3 — a 2-wins/1-tie/2-losses pattern over 5 regimes with small gaps and no variance.
2. **"~75% lower inference compute"** is derived from quadratic beam-width scaling, not measured. Should be reported as "~75% fewer beam expansions" or backed with wall-clock / FLOPs.

Several key ablations are missing: vanilla GRPO vs. BGRPO at matched compute; the rank-decay term vs. a step-function reward; seed variance. These don't invalidate the direction of the claims, but they soften the quantitative headline numbers.

**Requested Changes:**

### Critical

1. **Frontier LLM baseline.** Evaluate GPT-4 / Claude / Gemini / DeepSeek-Math zero-shot and few-shot on a held-out subset (100+ problems), plus at least one tool-augmented variant (LLM + SymPy or Mathematica). Without this, the paper cannot claim its small specialized model is competitive with anything. This is the single most important change.

2. **Classical-algorithm baseline.** Report Faugère–Perret or von zur Gathen's algorithm on the same synthetic test distributions (success rate + runtime). Even a partial comparison (univariate regime) would contextualize the neural results.

3. **Soften the Mathematica comparison** to "competitive with" unless Table 3 is substantially expanded with more regimes, variance, and a second simplification metric.

4. **Direct BGRPO ablation** at matched compute: (i) SFT, (ii) SFT + vanilla GRPO, (iii) SFT + BGRPO without rank signal, (iv) SFT + BGRPO with rank signal. Current Fig. 8 is missing (ii).

5. **Error bars / seed variance** on Figs. 1–8. Several comparisons turn on small gaps.

6. **Back the 75% compute claim** with wall-clock or FLOPs, or rephrase as "beam expansions."

### Strengthening

7. **Correctness criterion.** State explicitly whether a prediction is graded by exact tuple match or by functional equivalence (since $g \circ h$ is invariant under reparametrization).

8. **Test-set disjointness.** Clarify how test examples are ensured non-overlapping with the 1M–2M training set.

9. **Targeted fix for sign tokens** — token-type-weighted loss, two-stage decode, or constrained decoding. A negative result would still be informative.

10. **Multivariate distribution-adaptation.** $\mathcal{D}_3$ is univariate-only; extending it would test whether rapid adaptation holds where it matters.

11. **Larger-scale scaling point** (e.g., 12 layers / 1024 dim) to test whether the data-dependent scaling threshold saturates.

12. **Tokenization ablation** — the ~50% sign-token accuracy may be partly tokenization-induced.

13. **Monomial heads** — a head-ablation intervention would turn qualitative attention maps into a real interpretability claim.

14. **Release the synthetic data generator and evaluation harness** — the four-axis framework is the paper's most transferable contribution.

15. **Writing fixes.** Abstract's "non-linear latent pattern discovery" is vague — say "polynomial decomposition." Fig. 4's x-axis should read "Embedding Dimension." Several in-text citations miss years. Table 3's "competitive performances bolded" needs a stated threshold.

---

> ### Author Response · Authors · 2026-04-24
> **Response for Reviewer `NdVw`**
>
> We thank the reviewer; we ran both missing baselines and completed the BGRPO ablation.
>
> ### 1. Missing baselines
>
> > Frontier LLM baseline.
> > Classical-algorithm baseline.
>
> We tested both with BGRPO's verifier on $(v_o, v_s)$ pairs $(2,3), (3,2), (3,3), (3,4), (4,3)$. Neither solves the task.
> We implemented and ran RREFComPoly (Guo et al. 2026), a reformulation of MultiComPoly (Faugère & Perret, 2009) and the only classical algorithm used for multivariate decomposition. It expects homogeneous inputs with as many inner polynomials as variables (FDP_H; Section II-B). Our problems are not homogeneous (0/500); after homogenization, three of five regimes also fail this constraint. The algorithm returned 0/500 at 100 problems per regime.
> A zero-shot sweep at N=100/regime of five non-reasoning LLMs (Claude Haiku 4.5, Gemini 2.5 Flash, GPT-4.1 mini, GPT-4o mini, DeepSeek v3.1) peaked at 0.30 [0.22, 0.40] for Claude Haiku 4.5 on $(3,4)$; every other cell ≤ 0.09. We chose single-pass non-reasoning variants as the closest comparison to our model's single beam pass; reasoning-tier and tool-augmented multi-turn setups use orders more compute and were excluded.
>
> | Regime | Claude Haiku 4.5 | Gemini 2.5 Flash | GPT-4.1 mini | GPT-4o mini | DeepSeek v3.1 | Transformer (ours, beam=50) |
> |---|---|---|---|---|---|---|
> | (2,3) | 0.28 [0.20, 0.38] | 0.06 [0.03, 0.13] | 0.04 [0.02, 0.10] | 0.03 [0.01, 0.09] | 0.00 [0.00, 0.04] | **0.69** |
> | (3,2) | 0.06 [0.03, 0.13] | 0.09 [0.05, 0.16] | 0.03 [0.01, 0.09] | 0.01 [0.00, 0.05] | 0.00 [0.00, 0.04] | **0.15** |
> | (3,3) | 0.23 [0.16, 0.32] | 0.05 [0.02, 0.11] | 0.02 [0.01, 0.07] | 0.01 [0.00, 0.05] | 0.00 [0.00, 0.04] | **0.24** |
> | (3,4) | **0.30 [0.22, 0.40]** | 0.03 [0.01, 0.09] | 0.01 [0.00, 0.05] | 0.01 [0.00, 0.05] | 0.00 [0.00, 0.04] | 0.28 |
> | (4,3) | **0.20 [0.13, 0.29]** | 0.05 [0.02, 0.11] | 0.08 [0.04, 0.15] | 0.02 [0.01, 0.07] | 0.00 [0.00, 0.04] | 0.09 |
>
> ### 2. Self-made benchmark and metrics
>
> > Self-made benchmark.
>
> Train/test are independent and filtered for overlap. Grading: valid iff it recomposes exactly; same check across BGRPO training, inference, and baselines. Leaf count: within a few leaves of FullSimplify on every regime.
>
> ### 3. Mathematica softening
>
> > Soften the Mathematica comparison.
>
> We will replace "outperforming Mathematica" with "competitive with FullSimplify on leaf count". FullSimplify is a simplifier, not a decomposer; matching its leaf counts is a byproduct.
>
> ### 4. BGRPO ablation and the 75% compute claim
>
> > Direct BGRPO ablation at matched compute.
> > Back the 75% compute claim with FLOPs.
>
> For each d=256, d=512, d=768 6-layer SFT init ($\mathcal{D}_2$ sweep), we trained at matched compute (400 updates, 8 problems/step, beam 32, $\beta=0.01$, lr=1e-5) on 60 held-out problems.
>
> | SFT init | SFT only (beam 30) | + vanilla GRPO | + BGRPO (binary) | + BGRPO (rank-aware) |
> |---|---|---|---|---|
> | d=256 | 26.7 | 50.0 | 43.3 | **56.7** |
> | d=512 | 21.7 | 53.3 | 30.0 | **56.7** |
> | d=768 | 25.0 | 51.7 | 33.3 | **56.7** |
>
> 56.7 at all three sizes reflects a hard subset capping performance uniformly; we will analyze these in the final paper.
> These numbers improve on the submission: we disabled std normalization, giving $A_i = r_i - \bar{r}$ (paper Section 2.3, following Liu et al. 2025). Vanilla GRPO lifts beam-30 by 23 to 32 points; rank-aware adds 3 to 7 more. Without normalization, BGRPO outperforms vanilla GRPO (Table 4). These replace Fig. 8; error bars to follow.
> On the 200-problem greedy set, BGRPO-rank reaches 44.0, 44.0, 45.5 at d=256, d=512, d=768, exceeding SFT at 26.7, 21.7, 25.0 (beam 30): post-BGRPO greedy beats pre-BGRPO best beam.
> On 75%: we assumed quadratic scaling, but beam search is linear in $K$. At matched accuracy, beam 30 (pre) = beam 16 (post-BGRPO rank): $1 - 16/30 \approx$ **50% fewer FLOPs**, not 75%. We retract "~75% lower inference compute". With normalization off, FLOPs savings exceed raw beam reduction. Definitive FLOPs/wall-clock to follow.
>
> ### 5. Error bars and scaling
>
> > Error bars / seed variance on Figs. 1–8.
> > Larger-scale scaling point (12 layers / 1024 dim).
>
> Multi-seed reruns (BGRPO ablation, distribution adaptation) are underway; initial results confirm our findings, error bars in the final manuscript. Re-running earlier supervised figures across seeds is infeasible in the rebuttal window. We will train a 12-layer, 1024-dim model on the 2M corpus and add it to Fig. 4 when compute permits.
>
> ### 6. Clarifications and writing fixes
>
> Correctness: composition check (substitute, expand, compare; non-unique decompositions). Tokenization (digit-per-token, separate sign) was chosen after testing alternatives; we document these. We accept writing fixes (abstract, Figure 4 x-axis, citation years, Table 3 caption) and will note cryptanalytic applications in Broader Impact.
>
> We are open-sourcing the full pipeline (data generation, pretraining, BGRPO, multi-sampling/beam-search inference, evaluation, classical/LLM baselines).

---

### Decision · Action_Editor_hXDj · 2026-05-13

**Recommendation:** Accept with minor revision

**Audience:**

Yes

**Audience Explanation:**

Yes, the paper falls into the area of reinforcement learning and proposes improvements to beam search methodology. These are of interest to the TMLR audience for the technical contribution, even if analyzed benchmarks are somewhat limiting.

**Claims And Evidence:**

Yes

**Claims Explanation:**

The initial reviews had some concerns over empirical rigor, including e.g. better comparison to some baselines and inclusion of assessment across various random seeds. There were also some concerns that the claims surrounding the reduction of computational cost were not sufficiently supported and around the clarity of the writing wrt made claims.
These aspects were carefully addressed in the rebuttal and to the satisfaction of all three reviewers.
The action editor agrees with this assessment and requests for the final minor revision to incorporate the rebuttal suggestions for additional discussion, clarifications, and additional results (in particular the still outstanding multi-seed analysis for error bars)